# Surgical and Clinical Outcomes of Microvascular Decompression: A Comparative Study between Young and Elderly Patients

**DOI:** 10.3390/brainsci12091216

**Published:** 2022-09-09

**Authors:** Grazia Menna, Alessandro Rapisarda, Alessandro Izzo, Manuela D’Ercole, Quintino Giorgio D’Alessandris, Alessandro Olivi, Nicola Montano

**Affiliations:** 1Department of Neuroscience, Neurosurgery Section, Università Cattolica del Sacro Cuore, 00168 Rome, Italy; 2Department of Neurosurgery, Fondazione Policlinico Universitario Agostino Gemelli IRCCS, 00168 Rome, Italy

**Keywords:** trigeminal neuralgia, microvascular decompression, elderly, surgery

## Abstract

Microvascular decompression (MVD) is the only etiological technique for the treatment of trigeminal neuralgia (TN). Whilst there is a consensus MVD is likely effective regardless of age, the elderly population is thought to be more prone to have a higher rate of surgical complication, morbidity, and mortality. The main objective of our single-center, retrospective study was to analyze the surgical and clinical outcomes of MVD in TN elderly patients. From a surgical series of patients with TN who had undergone MVD from April 2018 to April 2022, 76 patients who matched the inclusion criteria were divided into two groups: twenty-five (32.9%) patients were older than 65 years and included in the elderly group, while the remaining fifty-one (61.1%) patients were below 65 years included in the non-elderly one. There were no differences between the groups in terms of acute pain relief (APR), Barrow Neurological Index (BNI) at follow-up, complications, and recurrence rate. In multivariate analysis (Cox proportional hazards regression analysis) the presence of an offending artery with nerve root distortion/indentation emerged as the only independent prognostic factor for pain-free survival (*p* = 0.0001). Our data endorse MVD as a safe and effective surgical procedure also for elderly patients with TN.

## 1. Introduction

Trigeminal neuralgia (TN) is defined as recurrent paroxysms of unilateral facial pain restricted to trigeminal distribution, lasting from a fraction of a second to 2 min, severe in intensity with an electric shock-like shooting, stabbing or sharp quality, and precipitated by innocuous stimuli [1]. The most common is the classical type, diagnosed when there is evidence of trigeminal neurovascular compression ipsilateral to the side of the pain. [2]. Depending on the pain characteristics, classical TN is further subdivided into a purely paroxysmal form (previously known as “typical” or “type 1”) and TN with concomitant continuous or near continuous pain occurs (previously known as “atypical” or “type 2”) [3,4,5]. Surgical treatments are reserved for patients with debilitating pain refractory to pharmacological treatments (currently consisting of carbamazepine and oxcarbazepine [3,6,7]. Of the available surgical intervention, microvascular decompression (MVD) is the only etiological treatment considered the first choice in classical TN: the surgical procedure has been extensively reported in the literature and its clinical efficacy has been widely recognized [8,9,10,11]. Whilst there is a consensus that MVD is likely effective regardless of age [12,13,14], the elderly population is thought to be more prone to having a higher rate of surgical complication, morbidity, and mortality. We believe this is of relevance because: (1) TN incidence increases with age; (2) the elderly are prone to report more side effects due to pharmacological therapy; (3) alternative therapies to surgery such as percutaneous balloon compression, often offered to elderly patients, are associated to poor long-term control. [13] The results available in the literature warrant new evidence on the topic.

Therefore, the aim of the present study was to analyze clinical and surgical outcomes in elderly patients (≥65 years) undergoing MVD surgery for TN compared to younger ones (<65 years), in an effort toward implementing tailored clinical decision-making on the topic.

## 2. Materials and Methods

### 2.1. Study Design and Patients′ Selection 

Clinical data were collected from patients diagnosed with classical TN according to the criteria of the ICHD-3 (1), who underwent MVD surgery at the Neurosurgical Department of Fondazione Policlinico Universitario Agostino Gemelli IRCCS (Rome, Italy) between April 2018 and April 2022.

All patients were operated on by the senior author (NM). Patients diagnosed with secondary trigeminal neuralgia (due to cerebellopontine angle tumors, AV-malformation, and multiple sclerosis) were a priori excluded.

All patients (regardless the age) with TN and a suspected neurovascular conflict on magnetic resonance imaging (MRI) were recommended to undergo MVD if pharmacological treatment was insufficient to control pain or significant side effects were reported. Patients without visible neurovascular conflict on MRI or bearing relevant comorbidities were recommended for alternative treatment (percutaneous balloon compression or gamma-knife radiosurgery). 

The patients who matched our inclusion criteria and underwent MVD surgery were divided into elderly and non-elderly groups, above or below 65 years, respectively. 

### 2.2. Outcome Data

For each patient, the following data were gathered: -Clinical pre-operative data: age, sex, affected side, TN type, pain duration (years), Barrow Neurological Institute Pain Intensity (BNI) score [15], previous surgery; -Intraoperative data: craniotomy size; use of neuronavigation during surgery [16]; surgical duration and mastoid opening. Intraoperative conflict was classified as: “contact only” when the offending artery was in contact with the nerve root but without any visible indentation and “distortion and/or indentation” when there was a distortion and/or an indentation of the nerve root caused by the offending artery [17,18];-Post-operative/follow-up data: acute pain relief (APR) (pain-free at hospital discharge); cerebrospinal fluid (CSF) leak after surgery and complication other than CSF leak; length of stay; pain-free survival determined at the most recent follow-up visit, need for re-operation and BNI at follow-up.

### 2.3. Statistical Analysis

Comparison of categorical variables was performed by means of the chi-squared statistic, using the Fisher exact test. Statistical comparison of continuous variables and ordinal variables was performed by the Student T-test and by the Wilcoxon signed-rank test, as appropriate. Long-term outcome was evaluated by Kaplan–Meier analysis with log-rank testing to compare the pain-free interval between groups. Cox logistic regression analysis was employed to analyze factors associated with recurrence. Differences were considered statistically significant at *p* < 0.05. Statistical analyses were conducted using Stat View version 5 software (SAS Institute, Inc, Cary, NC, USA).

### 2.4. Ethical Approval

Ethical approval was waived by the local Ethics Committee in view of the retrospective nature of the study and all the procedures being performed were part of routine care. Informed consent was obtained from all individual participants included in the study.

## 3. Results

### 3.1. Patients Characteristics

The sample studied consisted of 76 patients with a mean age of 59.6 ± 9.79 years. Twenty-five (32.9%) patients were included in the elderly group (≥65 years); the remaining fifty-one (61.1%) patients were included in the non-elderly group (<65 years). The two groups were not significantly different in terms of clinical pre-operative data (Table 1). 

Intraoperative data were similar between non-elderly and elderly patients as well. In particular, this was true in terms of craniotomy size, use of neuronavigation, surgical duration, mastoid opening, and intraoperative conflict type (see Table 2). 

### 3.2. Outcome Data

Outcome data are summarized in Table 3. Briefly, there was no difference in the length of hospital stay and in the complication rate between the two groups. Overall, we obtained an APR in 75 out of 76 cases (98.6%) with no difference between the two groups as well. BNI at follow-up was significantly reduced compared to pre-operative one (*p* < 0.0001; 1.40 ± 0.96 and 4.13 ± 0.71, respectively), with no difference between the two groups. Within the follow-up period, no significant difference was found in terms of recurrence between the two age groups (Table 3; Figure 1). Patients with recurrent pain were treated with drugs with no need for reoperation in any case. Only one case of CSF leak required a reoperation. The complications other than the CSF leak were the following: slight hemiparesis due to pontine ischemia completely recovered within 2 months; transient diplopia recovered in 1 week; occurrence after 1 month of ipsilateral frontoparietal chronic subdural hematoma treated with burr holes; transient hearing loss completely recovered in 2 weeks and a case of hypoesthesia recovered in 3 weeks.

### 3.3. Risk Factor Analysis for Pain Recurrence 

At univariate analysis (Kaplan–Meier survival analysis) no difference in pain-free survival was found when stratifying patients for age (*p* = 0.432; Figure 1), sex (*p* = 0.857), post-operative complications occurrence (*p* = 0.507) and evidence of previous surgery (*p* = 0.573). The presence of a trigeminal nerve root distortion/indentation was associated with a statistically significant higher pain-free survival (*p* < 0.0001; Figure 2).

A multivariate survival model for pain-free survival (Cox proportional hazards regression analysis) including age, sex, evidence of previous surgery, post-operative complications occurrence, and intraoperative conflict type (Table 4). The presence of an offending artery with nerve root distortion/indentation emerged as the only independent prognostic factor for pain-free survival (*p* = 0.0001; Table 4). 

## 4. Discussion

TN is the most frequently reported type of facial pain, with an overall incidence ranging from 12.6/100.000/year to 27/100.000/year [19]. Classic TN has been related to neurovascular compression in the prepontine cistern at the nerve root entry zone owing to the presence of an abnormal artery or vein. Demyelination in the region of nerve compression, activation of peripheral receptors, transmission, and convergence of nociceptive information onto common central nervous neurons, generation of spontaneous nerve impulses, and their ephaptic transmission to adjacent fibers have been advocated to play a role in the pathogenesis of the disease [20]. Radiosurgery and percutaneous destructive techniques are the available surgical minimally invasive alternatives for refractory TN. However, those techniques are often associated with a higher recurrence rate at follow-up. MVD is known to be the only etiological surgical intervention since its primary aim is the resolution of the conflict between the nerve at its root entry zone and the offending vessel. Even if MVD has been performed for medically refractory TN regardless of age, there is the idea that this type of surgery is less safe for the elderly, with a higher number of complications and a greater risk of recurrence [21,22,23,24]. Thus, more often TN elderly patients are approached with palliative procedures which are considered safer in this population. 

### 4.1. Comparative Analysis between Elderly and Non-Elderly Group

The herein presented data confirm what has already been reported in the literature. [25,26,27]. Even though the median length of stay was slightly longer in the elderly group, an excellent outcome was obtained for almost all patients: in our series, 75 patients (98.6%) reported an APR (98% in the non-elderly group vs. 100% in the elderly group) with a good pain outcome at follow-up and low complication rate not different between the two groups. Most importantly, we found no difference in terms of recurrence between the elderly and non-elderly groups, as previously demonstrated [13]. This suggests elderly patients can be safely referred to MVD surgery with excellent early and long-term pain outcomes and without an increased risk of recurrence. 

### 4.2. Risk Factor Analysis for Pain Recurrence

As is widely known, the key to MVD is to identify the offending vessel and to achieve effective decompression of the nerve. The offending vessel is most commonly an artery rather than a vein [28]. We studied different factors (namely age, sex, post-operative complications occurrence, evidence of previous surgery, and the kind of neurovascular conflict) as possible prognosticators of recurrence after MVD and found that the presence of an artery only in contact with the nerve root but without any visible indentation was the only independent predictor of TN recurrence after successful MVD surgery. Our results agree with other studies, showing functional outcome is mainly related to clear identification and resolution of the arterial conflict [29,30]. A recent meta-analysis reinforced this idea. This work demonstrated that the duration of symptoms before surgery less than 5 years and the clear identification of the neurovascular conflict were positive prognostic factors [31]. These findings evidenced that the anatomical relationship between the artery and the nerve root is the main predictor for clinical outcome, confirming that elderly patients should not be excluded a priori from MVD.

### 4.3. Limitations 

The relatively low number of patients in the elderly group and the retrospective nature of data are the main limitations of our study.

Another critical issue could be the cut-off for the differentiation of the two groups. In the absence of universally recognized criteria for discriminating against elderly/non-elderly patients, we relied substantially on the most frequently reported cut-off in the literature [32,33,34].

Furthermore, some general clinical data potentially affecting the outcome and the complication occurrence (such as comorbidities and ASA status) were not recorded and evaluated.

## 5. Conclusions

MVD surgery in elderly patients remains a controversial topic. In this single-center, retrospective study, we compared elderly and non-elderly patients treated with MVD in terms of clinical and surgical outcomes. We observed an excellent early and long-term pain outcome in both groups with no significant differences in terms of complications. The presence of a clear neurovascular conflict with nerve distortion/indentation was associated with a better long-term outcome. Our results endorse MVD as an effective and safe surgical procedure for elderly patients with TN.

## Figures and Tables

**Figure 1 brainsci-12-01216-f001:**
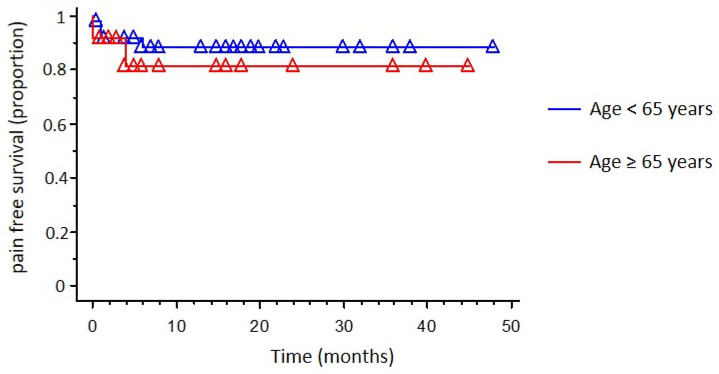
Kaplan–Meier cumulative probability of pain-free survival in patients submitted to MVD, stratified by age.

**Figure 2 brainsci-12-01216-f002:**
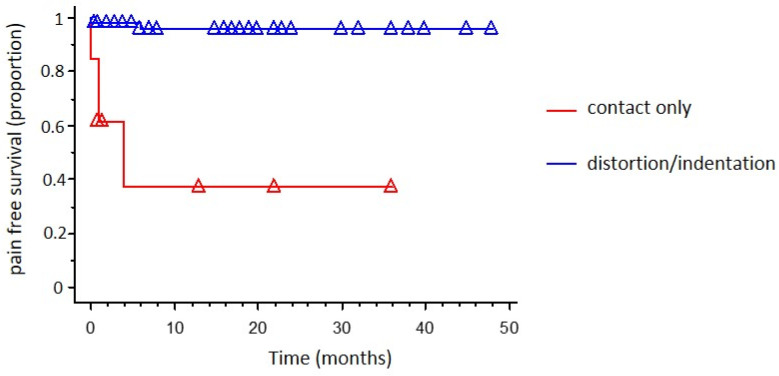
Kaplan–Meier cumulative probability of pain-free survival in patients submitted to MVD, stratified by kind of neurovascular conflict.

**Table 1 brainsci-12-01216-t001:** Patients′ demographics and clinical characteristics.

	ALL CASES N = 76	AGE < 65 N = 51	AGE ≥ 65 N = 25	*p* Value
**Sex (M)**	29 (38.1%)	22 (43.1%)	7 (28%)	0.201
**Side (L)**	29 (38.1%)	17 (33.3%)	12 (48%)	0.216
**Type (Classical TN)**	75 (98.6%)	51 (100%)	24 (96%)	0.151
**Pain duration (years)**	6.37 ± 4.63	6.36 ± 5.13	6.40 ± 3.55	0.977
**BNI pre-op**	4.13 ± 0.71	4.11 ± 0.65	4.16 ± 0.85	0.811
**Previous surgery**	6 (7.9%)	5 (9.8%)	1 (4%)	0.657

Abbreviation: BNI Barrow Neurological index, M male, L left, TN trigeminal neuralgia.

**Table 2 brainsci-12-01216-t002:** Intraoperative data comparison.

	ALL CASES N = 76	AGE < 65 N = 51	AGE ≥ 65 N = 25	*p* Value
**Craniotomy size (cm^2^)**	3.60 ± 1.44	3.65 ± 1.53	3.51 ± 1.26	0.696
**Neuronavigation (Yes)**	40 (52.6%)	30 (58.8%)	10 (40%)	0.164
**Surgical duration (min)**	116.34 ± 24.98	116.86 ± 27.42	115.28 ± 19.53	0.797
**Mastoid opening (Yes)**	13 (17.1%)	9 (17.6%)	4 (16%)	0.857
**Double Conflict**	6 (7.8%)	5 (9.8%)	1 (4%)	0.657
**Nerve root distortion/indentation**	63 (82.8%)	43 (84.3%)	20 (80%)	0.748

**Table 3 brainsci-12-01216-t003:** Outcome data comparison.

	ALL CASES N = 76	AGE < 65 N = 51	AGE ≥ 65 N = 25	*p* Value
**APR (Yes)**	75 (98.6%)	50 (98%)	25 (100%)	0.481
**CSF leak**	4 (5.2%)	2 (3.9%)	2 (8%)	0.594
**Complications other than CSF leak**	5 (6.5%)	3 (5.8%)	2 (8%)	0.534
**Length of stay (days)**	5.09 ± 1.8	5 ± 1.68	5.28 ± 2.05	0.528
**Median follow-up (range 3–48 months)**	13.5 ± 11.99	12.47 ± 11.93	15.60 ± 12.06	0.288
**BNI at follow-up**	1.40 ± 0.96	1.35 ± 0.86	1.52 ± 1.15	0.483
**Recurrence**	9 (11.84%)	5 (9.8%)	4 (16%)	0.432

Abbreviation: APR acute pain relief, BNI Barrow Neurological index, CSF cerebrospinal fluid, N number.

**Table 4 brainsci-12-01216-t004:** Multivariate analysis for pain-free survival.

	Coef	Coef/SE	Chi-Square	*p* Value	HR	95% Lower	95% Upper
**Age**	−0.592	−0.795	0.632	0.426	0.55	0.129	2.380
**Sex**	−1.438	−1.642	2.69	0.1	0.23	0.043	1.322
**Previous surgery**	−0.45	−0.388	0.15	0.69	0.63	0.064	6.305
**Post-op complications**	−1.02	−0.953	0.909	0.34	0.33	0.034	3.202
**Intraoperative conflict type**	3.781	3.89	15.132	0.0001	43.86	6.52	294.83

Abbreviations: Coeff coefficient, HR hazard ratio, SE standard error.

## Data Availability

Not applicable.

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
