# Peer review of "Surgical and Clinical Outcomes of Microvascular Decompression: A Comparative Study between Young and Elderly Patients"

_brainsci, 2022, doi:10.3390/brainsci12091216_

Round 1

Reviewer 1 Report

The authors investigated surgical and clinical outcomes of microvascular decompression in elderly patients. They report that microvascular decompression is safe and suitable for elderly patients. 

I have one major comment and two minor comments that authors might find helpful.   Major 1. How did the authors select the age parameters? They took above 65 years as elderly group and below that as non-elderly group. What was the basis of choosing the age of 65 years? Why they chose the age 65 years should be mentioned in the manuscript.   Minor 1. In the introduction section, the authors described mainly trigeminal neuralgia and its cause and classification-related information, which are already well-known and do not need to be described elaborately. The introduction section should contain more background information regarding  why elder people were thought to be more prone to complications and why they should not be because this is their main point to defend.   2. Last line of point 4.1 "this is of extreme ........... in elderly as well" should be written in a more simple way. 

Author Response

Dear Reviewer,

We thank you for the comments on the paper and for your detailed review. We are glad that this manuscript was of interest to you. Following your carefully structured observations and recommendations, the manuscript was modified. Below is a point-to-point response to your comments.

  • The authors investigated surgical and clinical outcomes of microvascular decompression in elderly patients. They report that microvascular decompression is safe and suitable for elderly patients. I have one major comment and two minor comments that authors might find helpful.   Major 1. How did the authors select the age parameters? They took above 65 years as elderly group and below that as non-elderly group. What was the basis of choosing the age of 65 years? Why they chose the age 65 years should be mentioned in the manuscript.  

We thank the reviewer for pointing out this important aspect. In the absence of universally recognized criteria for discriminating elderly/non-elderly patients, we relied substantially on the most frequently reported cut-off in the literature, as reported by the following papers:

  • Schär RT, Tashi S, Branca M, et al. How safe are elective craniotomies in elderly patients in neurosurgery today? A prospective cohort study of 1452 consecutive cases. J Neurosurg. 2020;134(3):1113-1121.
  • Soleman J, Ullmann M, Greuter L, Ebel F, Guzman R. Mortality and Outcome in Elderly Patients Undergoing Emergent or Elective Cranial Surgery. World Neurosurg. 2021;146:e575-e589.
  • Johans SJ, Garst JR, Burkett DJ, et al. Identification of Preoperative and Intraoperative Risk Factors for Complications in the Elderly Undergoing Elective Craniotomy. World Neurosurg. 2017;107:216-225.

 This coincides with the one routinely used in our institution to differentiate the kind of treatment offered to the patient. The manuscript has been updated accordingly [Limitations section, Paragraph 4.3]

  • Minor 1. In the introduction section, the authors described mainly trigeminal neuralgia and its cause and classification-related information, which are already well-known and do not need to be described elaborately. The introduction section should contain more background information regarding why elder people were thought to be more prone to complications and why they should not be because this is their main point to defend.  

We thank the reviewer for the thoughtful comment. The introduction has been shortened and rewritten focusing on the elderly undergoing MVD which, as correctly written, is the core of our paper. [Introduction section]

  • Last line of point 4.1 "this is of extreme ........... in elderly as well" should be written in a more simple way. 

Sentence was rephrased as requested.

Reviewer 2 Report

The authors studied the outcomes of microvascular decompression in elderly patients

The “ Surgical and clinical outcomes of microvascular decompression in elderly patients ” have to be revised because in the paper the number of the non-elderly is more of the elderly/which may put  comparison …

The introduction is relatively long and is to be compressed

Numbers at the beginning of the sentence must be written in letters ( example 25 / line 23 )

The characterization section should be described with more detail.

tables must be readable in themselves (table 1 / table 3 ) : abbreviations must be put at the bottom of the tables

Have you checked the repetitiveness of the results, especially for outcomes data

The title of figures (1 and 2) is to be checked; of the results that appear on the title ?.

the unit must be put that is for the axis abscissa that ordinate ( time in ? pain free survival in ?)

Avoid mentioning and describing Methods  in the discussion  (lines 173-175)

A validation/verification of the numerical model is to be performed.

The English level is to be improved.

Author Response

Dear Reviewer,

We thank you for the comments on the paper and for your detailed review. We are glad that this manuscript was of interest to you. Following your carefully structured observations and recommendations, the manuscript was modified. Below is a point-to-point response to your comments.

The authors studied the outcomes of microvascular decompression in elderly patients

  • The “ Surgical and clinical outcomes of microvascular decompression in elderly patients ” have to be revised because in the paper the number of the non-elderly is more of the elderly/which may put  comparison …

Thank you for your comment. We think that the title of this paper can be considered appropriate. The aim of this study was to report the outcomes of MVD in elderly patients. We had two choices: 1) to report a surgical series of elderly patients (no control group); 2) to perform, as we did, a comparative study with younger patients to enhance the power of our work. We agree that elderly patients are lower in number but this is the consequence of the fact that MVD is offered more frequently to younger patients. The conclusion of our study is that MVD is effective and safe for elderly patients with TN and can be offered to these patients  .

  • The introduction is relatively long and is to be 

We thank the reviewer for the thoughtful comment. The introduction has been shortened and rewritten focusing on  elderly patients undergoing MVD.

  • Numbers at the beginning of the sentence must be written in letters example 25 / line 23 )

We thank the reviewer for the comment. The typo was corrected.

  • The characterization section should be described with more detail.

The inclusion criteria for surgery have been updated and are now detailed in Methods section, as required.[Methods section, Paragraph 2.1]

  • Tables must be readable in themselves (table 1 / table 3 ) : abbreviations must be put at the bottom of the tables

Tables have been updated as requested.

  • Have you checked the repetitiveness of the results, especially for outcomes data

Thank you for your comment. We carefully checked the results section.

  • The title of figures (1 and 2) is to be checked; of the results that appear on the title?

Figures 1 and 2 have been updated as requested.

  • The unit must be put that is for the axis abscissa that ordinate ( time in ? pain free survival in ?)

Updated version of Figures 1 and 2 are enclosed in the manuscript

Time (months); Pain free survival (proportion)

  • Avoid mentioning and describing Methods in the discussion  (lines 173-175)

Manuscript has been updated and the reference removed.

  • A validation/verification of the numerical model is to be performed.

Thank you for your comment. This is a retrospective study. A formal sample size analysis (as for prospective studies) is not possible. We reported this in the limitation section: “the relatively low number of patients in the elderly group and the retrospective nature of data are the main limitations of our study”

  • The English level is to be improved.

Manuscript has been carefully reviewed, as recommended.

Round 2

Reviewer 2 Report

The paper can be accepted after minor revision:

The title  “ Surgical and clinical outcomes of microvascular decompression in elderly patients ” have to be revised because in the paper the number of the non-elderly is more of the elderly/  which may put  comparison …

 reference 4 has been deleted (line 60: 4 crossed out) but still exists in the manuscript

Numbers at the beginning of the sentence must be written in letters ( example 25 / line 23 )

tables must be readable in themselves (table 1 :  N , M, L/ table 3: N / table 4: Coef/SE

, HR ) : abbreviations must be put at the bottom of the tables

 Avoid mentioning and describing Methods  in the discussion  (lines 220-224)

Author Response

Dear Editor and Reviewers,

We thank you for the comments on the paper and for your detailed review. We are glad that this manuscript was of interest to you. Following your carefully structured observations and recommendations, the manuscript was modified. Below is a point-to-point response to your comments.

REVIEWER 2

The paper can be accepted after minor revision:

The title  “ Surgical and clinical outcomes of microvascular decompression in elderly patients ” have to be revised because in the paper the number of the non-elderly is more of the elderly/  which may put  comparison …

We thank the reviewer for the comment. We rephrased the title as follows: “Surgical and clinical outcomes of microvascular decompression. A comparative study between young and elderly patients”

 reference 4 has been deleted (line 60: 4 crossed out) but still exists in the manuscript.

The typo has been corrected.

Numbers at the beginning of the sentence must be written in letters ( example 25 / line 23 )

The typos have been corrected.

tables must be readable in themselves (table 1 :  N , M, L/ table 3: N / table 4: Coef/SE, HR ) : abbreviations must be put at the bottom of the tables

The proper abbreviations have been updated

 Avoid mentioning and describing Methods  in the discussion  (lines 220-224)

Sentence has been rephrased as suggested.